# Ultra-Short-Pulse Laser Welding of Glass to Metal with a Shear Strength Above 50 MPa

**DOI:** 10.3390/mi16050538

**Published:** 2025-04-30

**Authors:** Lukas Günther, Jens Ulrich Thomas, Jens Hermann, Axel Ohlinger, Dominique de Ligny

**Affiliations:** 1Department of Materials Science and Engineering, Institut für Glas und Keramik, Friedrich-Alexander-Universitat Erlangen-Nürnberg (FAU), Martensstrasse 5, 91058 Erlangen, Germany; 2SCHOTT AG, Hattenbergstraße 10, 55122 Mainz, Germany; jensulrich.thomas@schott.com (J.U.T.);

**Keywords:** laser, ultra-short-pulse, welding, glass, metal, bonding

## Abstract

We report an ultra-short-pulse laser welding process that allows one to consistently weld Borofloat^®^ 33 glass to aluminum with a shear strength above 50 MPa. We explored the morphology of the welding seam and quantified the quality of the bonding by statistically determining the shear strength with more than 30 samples. The results of the shear strength tests indicate that the intrinsic shear strength of the aluminum serves as the upper limit of the glass-to-metal bond.

## 1. Introduction

Bonding of glasses to metal is indispensable for technical applications in many areas, such as the mounting of optical components, medical devices, micro electronics, or hermetic view ports. However, due to the brittle nature of glass, a mechanical connection is often challenging, and a metal frame is necessary. Established processes, such as anodic or adhesive bonding, have disadvantages that often exclude their utilization. Adhesives can cause part movement during the curing process, and later aging processes of the adhesive are often an issue. For this reason, this process is often not suitable for high-precision optics or vacuum applications. Anodic bonding enables a direct bond, but the glass and metal must be heated for the process. This not only causes problems due to the difference in the coefficient of thermal expansion of glass and metal but also often excludes the application for micro-electronic devices. Direct welding of glass to metal using ultra-short-pulse (USP) laser radiation offers considerable advantages. The lack of additional adhesives and the localized heat-affected zone (HAZ) offer a significant improvement.

USP laser welding of glass with copper was first demonstrated in 2008 by Osaka et al. [1] using a femtosecond laser, achieving a shear strength of 17 MPa. In 2014, Carter et al. [2] demonstrated the possibility of successfully welding various glasses with various metals, including steel, copper, and aluminum. In the following years, several publications investigated different aspects of USP laser welding. Guodong [3] highlighted a process without pressure assistance with a shear strength of 2.34 MPa. Further studies of USP laser welding of SiO_2_ and BK7 to aluminum by Carter et al. [4] showed 13 MPa strength and the ability to withstand thermal fluctuations, which is essential for industrial applications. In 2018, Matsuyoshi et al. [5] achieved femtosecond welding of glass to rough copper ($R_{z}=7.8\;\mu m$ and $R_{a}=0.21\;\mu m$), resulting in a shear strength of 1 MPa. The use of USP laser welding for epoxy-free optical instruments was carried out by Lafon et al. [6] for several glasses and metal combinations. In the following year (2021), Wang et al. [7] reported a shear strength of 8.79 MPa for the welding of soda-lime glass to stainless steel, and Li et al. [8] reported 10.9 MPa for the welding of Borofloat^®^ 33 (Schott AG, Mainz, Germany) to copper. For an industrial application of the process, the process time and shear strength parameters are of crucial importance. Previous studies [1,2,3,4,6,7,8] were comparatively slow, with a feed rate of ≤$1\;mm\;s^{-1}$ and shear strengths of less than 20 MPa, which are not high enough for key applications. In this study, we present welding results with a feed rate of $10\;mm\;s^{-1}$ and average shear strengths greater than 50 MPa. Laser conditions were chosen to form a liquid zone localized at the metal–glass interface where both components would mix.

## 2. Experimental Procedure

### 2.1. Welding Setup

The setup used in this study is shown in Figure 1. We used an ultra-short-pulse laser (Tangor, Amplitude, Pessac, France), emitting pulses with a pulse duration of τ=3 ps ($sech^{2}$-Fit) and wavelength of λ=1030 nm. The 1/$e^{2}$ beam diameter $d_{\mathrm{beam}}$ of the unfocused laser beam was 3.3 mm. In order to control the output power of the laser, a λ/2 waveplate and a polarization beam splitter were inserted into the beam path.

The laser beam is focused by a long-distance microscope objective (Mitutoyo, Kawasaki, Japan) with a numerical aperture (NA) of 0.26, resulting in a (1/$e^{2}$) spot diameter ($d_{\mathrm{spot}}$) of 8 μm in the focal plane. A chromatic beam splitter allows the use of the same objective for alignment and process control. The imaging arm of the system contains a tube lens (Mitutoyo, Kawasaki, Japan) with f=200 mm and a CCD camera (Allied Vision, Stadtroda, Germany). The focusing and imaging parts of the setup are mounted on the Z-axis for precise alignment and height control during the welding process within the depth of field of ±4.1 μm.

The welding samples are held by a clamping mechanism, which keeps the parts in close contact before and during welding. It is mounted on an X-Y motorized stage, which enables continuous movement of the samples during the welding process.

### 2.2. Materials, Preparation, and Welding

We use conventional 20 μm thick aluminum foil and 1 mm thick Borofloat^®^ 33 (SCHOTT AG) glass slides. For the welding process, the distance between the materials is particularly important to consider. Since the thin aluminum foil can be pressed against the glass, it is not the waviness but the roughness of the two materials that determines their spacing. White-light interferometry was used to determine the roughness of the aluminum foil at $R_{z}=2.6\;\mu m$ and $R_{a}=0.2\;\mu m$, which is significantly greater than the roughness of the fire-polished glass surface with 15 nm [9]. Furthermore, a dust-free surface is necessary for a successful welding process, which is why the samples were cleaned with compressed air and then brought into contact.

In order to operate in the heat accumulation regime during the welding process, the repetition rate *R* of the USP laser was set to 500 kHz. Motivated by the studies of Richter et al. [10] and Hecker et al. [11], we decided to use a pulse burst to reduce cracks and weld line disruptions in the glass. Each pulse is split into four sub-pulses, separated by 25 ns. Figure 2 shows a schematic of the burst mode used during the welding process. Each sub-pulse has a pulse energy $E_{p}$ of 0.43 μJ measured after the objective. During the welding process, the feed rate is kept constant at $10\;mm\;s^{-1}$. The glass–metal interface is in line with the focal plane of the objective, resulting in a fluence of(1)$$\phi =\frac{E_{p}}{\pi \;\cdot \;{\left(\frac{d_{\mathrm{spot}}}{2}\right)}^{2}}=0.86\;J/cm^{-2}$$
at the interface. The process parameters are summarized in Table 1.

## 3. Results

### 3.1. Characterization of the Weld Lines

The welded glass–aluminum joints produced in this study show a characteristic morphology. In Figure 3a, a cross section of the weld seam between the aluminum foil and the Borofloat^®^ 33 (SCHOTT AG, Mainz, Germany) glass slide is shown. The picture was taken with a scanning electron microscope (SEM) (Zeiss Sigma 500, Oberkochen, Germany). Due to the thermal influence of the USP laser, a heat-affected zone (HAZ) at the glass–metal interface was formed and defines the weld seam to join the two materials. This weld seam is characterized by an intermixing zone in which both aluminum and components of the glass are present. We indicate the top and bottom of the intermixing zone with two horizontal dashed lines. While the boundary of the intermixing zone with the glass is sharp and clearly visible in the SEM cross-section, its interface to the aluminum is more gradual, extending to ca. 1 μm. Therefore, we also conducted an elemental analysis using the energy-dispersive X-ray spectroscopy (EDX) feature of the SEM. The relative intensities of silicon and aluminum were measured vertically from the top of the weld line into the aluminum foil along the double arrow plotted in Figure 3a. The EDX intensities are normalized to the bulk intensities of aluminum and silicon. Following the line scan from the bottom to the top, in the lower part, mainly aluminum is clearly detectable in the spectrum. The amount of silicon increases sharply at a thickness of 1 μm. Above the green dashed line, in the mixing zone, the proportion of silicon increases while the proportion of aluminum decreases. The upper limit represents the area where the concentration of aluminum drops and only the components of the raw glass are detectable. The height of this intermixing zone is 6–7 μm, and the majority of the welding seam is located in the glass volume. The depth of penetration into the metal layer is less than 3 μm.

### 3.2. Shear Strength Test

As a test sample configuration, we chose two $5\times 5\times 1\;{\mathrm{mm}}^{3}$ glass slides with aluminum foil sandwiched in between. The total thickness in the range of 2–2.1 mm is given by the slides and the aluminum foil thickness. A sketch of the sample design is given in Figure 4. The penetration depth of the weld seam is less than 3 μm into the aluminum foil. Consequently, the weld seams on each side of the foil do not exceed the 20 μm thickness of the aluminum foil. This ensures that the welded regions from opposite sides remain separated by the aluminum layer. Consequently, there is no direct glass–glass bonding or diffusion across the aluminum layer, and the aluminum layer functions as a continuous barrier throughout the joint. This enables the glass–metal weld seams on both sides of the foil to be considered as separate joints. The aluminum foil is welded in a cross pattern consisting of 11 vertical and 11 horizontal lines. The lines have a line spacing of 100 μm, are 5 mm long, and extend over the entire length of the glass sheet. In the manufacturing process, the two glass sheets were welded one after the other, and the samples were flipped between the two process steps so that the laser was always focused through the glass onto the glass–metal interface.

The welding area *A* of each glass–aluminum interface can be calculated using the 8 μm weld line width and the number of welding lines. This leads to(2)$$A=22\;\cdot \;\left(8\;\mu m\;\cdot \;5\;mm\right)-\frac{121}{2}\;\cdot \;8\;\mu m\;\cdot \;8\;\mu m=0.872\;mm^{2},$$
considering that the crossing points of the lines are not counted twice.

For a quantitative investigation of the weld seams, a shear strength test was performed. We used a custom-built groove holder with a tip-tilt mechanism and a force testing system (Instron 5969, Norwood, MA, USA) with a 100 N load cell. The ball bearing minimizes the pull force during the test and allows one to predominantly measure the shear strength of the sample. A sketch of the test setup can be seen in Figure 5. Following the recommendation of ISO 13445 [12], the test is performed horizontally at a constant feed rate of $1.5\;\mathrm{mm}\min^{-1}$. The shear plane lies in the center of the two groove holders, corresponding to the metal film layer of the symmetrical samples. The test is terminated when the specimen breaks, i.e., when the force drops sharply. The maximum value of the compression force reached is recorded.

In total, 32 samples underwent the destructive shear strength test. After the test, the interfaces of the glass sheet and metal foil that were connected are imaged. The samples show a characteristic fracture pattern. This can be seen in Figure 6. If the welded joint fails, the joint is weakened on the corresponding sides. As a result, the joint between the metal and glass breaks open. This is clearly recognizable in Figure 6a. The metal foil is still welded to the metal piece, while in Figure 6b, it can be seen that the bond has broken. This indicates that as soon as the weld seam on one side of the foil begins to fail, the connection on the same side breaks. For this reason, it is justified to consider the surface welded to the respective glass slide separately for the shear strength calculation. As an example, the measurement curve of one of the 32 samples is plotted in Figure 7. During the test, the shear force increases over time because of the constant feed rate. After the maximum shear force of the sample is reached, the sample breaks apart. For this sample, the maximum force is 67.21 N. By dividing the force with the welding area $A=0.872\;mm^{2}$ (Equation (Equation 2)), the shear strength τ of this sample amounts to(3)$$\tau =\frac{F}{A}=\frac{67.21\;N}{0.872\;mm^{2}}=77.08\mathrm{MPa}.$$

This destructive shear test was performed on all 32 samples, and the shear strength of each sample was determined using Equation (Equation 3). The results are summarized in Figure 8 as a double logarithmic Weibull plot. For the measured data, the Weibull cumulative distribution function(4)$$G\left(\tau \right)=1-e^{-{\left(\frac{\tau }{\alpha }\right)}^{\beta }}$$
was fitted, and the form parameter α=64.5 and the shape parameter β=3.1 were determined. Therefore, the mean value μ of the Weibull distribution is(5)$$\mu =\alpha \;\cdot \;\Gamma \left(\frac{1}{\beta }+1\right)=57.7\mathrm{MPa}$$
and the standard deviation is(6)$$\sigma =\sqrt{\alpha^{2}\;\cdot \;\left(\Gamma \left(\frac{2}{\beta }+1\right)-\Gamma^{2}\left(\frac{1}{\beta }+1\right)\right)}=20.6\mathrm{MPa}.$$

The measured values show good agreement with the Weibull distribution for a shear resistance below 70 MPa, which indicates that the typical failure process of glass is the cause of the bond failure. An asymptotic behavior is present for the measured values above 70 MPa, and a strength of about 90 MPa seems to be the upper limit of the shear strength. The highest measured value is 91.3 MPa. This asymptotic behavior and the associated deviation from the Weibull distribution indicates that there is an additional failure process.

A more detailed analysis of the fracture behavior of the welded side of the glass provides information about the cause of the failure of the joint during the shear strength test. An overview of the fracture patterns can be seen in Figure 8b,c. In these microscope images, the intermixing zone is recognizable as black residual material in the welding lines. The fracture pattern in Figure 8c shows two characteristics along the welding seam. In some parts, the intermixing zone of the weld is still connected to the glass, while in other parts, it has broken out of the bulk glass. This indicates that the mixing zone and the transition from the mixing zone to the glass are the main weak points in these areas. The sample in Figure 8c withstood a shear strength of 34.71 MPa. All samples that are outlined with the red rectangle in Figure 8a show a similar fracture behavior. This leads to the conclusion that the bond between the intermixing zone and the glass, or the intermixing zone itself, is the weak point for these samples. This is, furthermore, consistent with the good agreement of the measured values with the Weibull distribution.

The analysis of the fracture patterns with a shear strength greater than 70 MPa, marked with a green rectangle in Figure 8a, shows a further characteristic. A microscope image of the sample that broke at a shear strength of 73.56 MPa is shown in Figure 8b. Along the welded seams, remnants of the aluminum foil can be seen. These were pulled out of the metal foil and are still connected to the glass slide after the shear test. In these cases, it was not the intermixing zone or the transition from the intermixing zone to the glass that was the weakest point of the joint but the aluminum foil. This additional fracture behavior explains the deviation from the equilibrium line of the Weibull plot to an asymptotic behavior. Indeed, the shear strength of pure aluminum is 90 MPa [13].

In contrast to previous work, we did not lap and polish the metal but used a foil instead, which was pressed between two glass sheets. With this approach, we could maximize the contact area. This is not achievable with lapping or polishing, since such a process produces a smooth but bowed surface [2]. Indeed, we now assume that the deciding factor for a successful glass-to-metal bond is not the roughness of the metal surface but solely its waviness.

Our approach also allowed for a large number of samples of the same quality, which is fundamental for laser parameter optimization. With the chosen laser parameters, we obtained weld lines with fewer cracks and disruptions and almost no stress birefringence in their vicinity. However, a detailed investigation of the stress birefringence surrounding the weld line is beyond the scope of this paper.

We believe that both factors—low waviness and the quality of the weld line—contribute to the high shear strength.

## 4. Conclusions

In this study, we successfully demonstrated the feasibility of USP laser welding of Borofloat^®^ 33 glass to aluminum, achieving shear strengths consistently exceeding 50 MPa. Our experiments revealed that the welding process, utilizing a picosecond laser, can produce strong and reliable joints with a feed rate of $10\;mm\;s^{-1}$ and without the need for extensive surface preparation or cleaning. The shear strength tests showed a mean strength of 57.7 ± 20.6 MPa, with the highest recorded strength reaching 91.3 MPa. These results indicate that the intrinsic shear strength of the aluminum foil serves as the upper limit of our bonding system.

The overall performance of the welded joints is promising, suggesting that USP laser welding is a viable technique for joining glass to metal in applications requiring high mechanical strength. Future work should focus on optimizing the welding parameters and exploring the long-term durability of joints under various environmental conditions. 

## Figures and Tables

**Figure 1 micromachines-16-00538-f001:**
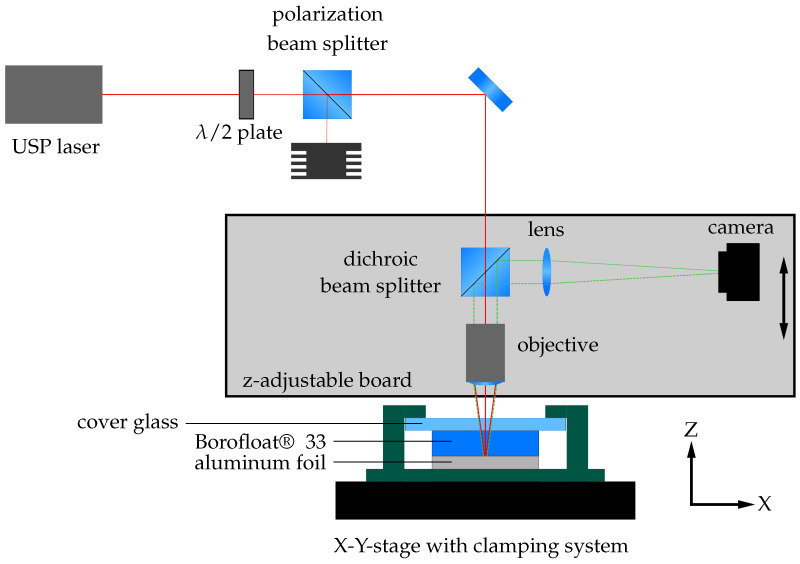
Experimental setup for the sample preparation. The ultra-short-pulse laser beam is focused by the optical setup at the glass–metal interface of the sample. A clamping device including a cover glass ensures that the sample glass to be welded and the aluminium foil are in close contact during the process. The device is fixed to an X-Y axis table, which allows constant movement during the process. A camera system within the optical path enables alignment and process control.

**Figure 2 micromachines-16-00538-f002:**
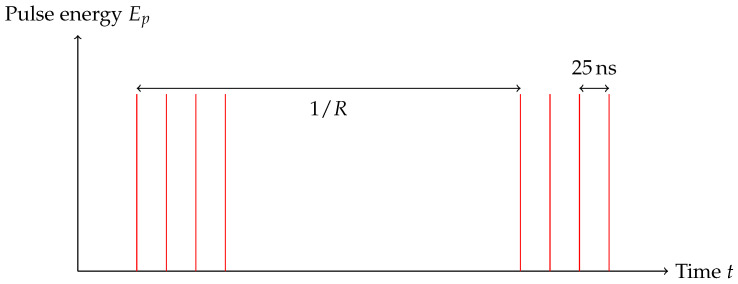
Schematic representation of the pulse regime used for the welding process. Each pulse contains four individual laser pulses spaced by 25 ns, with a pulse energy of 0.43 μJ each.

**Figure 3 micromachines-16-00538-f003:**
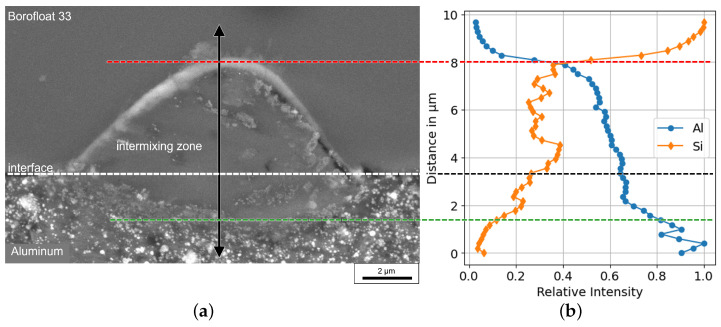
Cross-section of a glass-to-metal weld seam. In (**a**), a SEM picture of the welding seam, characterized by the intermixing zone of glass and aluminum, can be seen. Through the center of this picture, an EDX line scan (black arrow) was performed to detect the concentration of silicon and aluminum. The data of this scan are plotted in (**b**). The spot diameter of the scan is ∼1.5 μm.

**Figure 4 micromachines-16-00538-f004:**
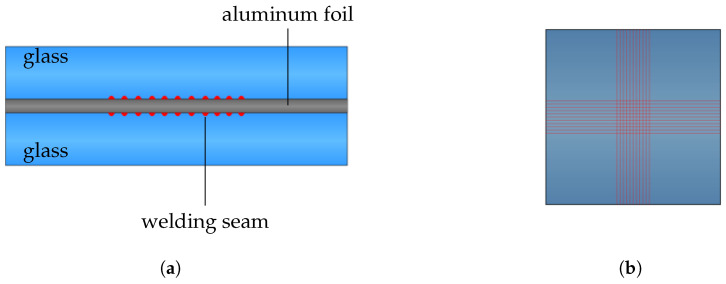
Sketch of the side view (**a**) and top view (**b**) of the welding samples made from two $5\times 5\times 1\;{\mathrm{mm}}^{3}$ glass sheets and a 20 μm thick aluminum foil. The welding pattern contains eleven vertical and horizontal welding lines with a length of 5 mm, shown as red lines, on each side of the aluminum foil. The line spacing between the welding seams is 100 μm and the pattern is centered on the glass sheet.

**Figure 5 micromachines-16-00538-f005:**
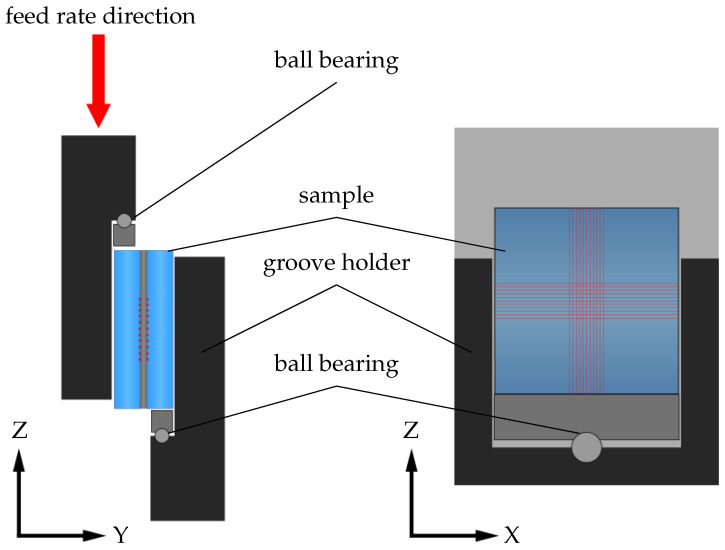
Sketch of the side and front view of the groove holder used to test the welding samples. The sample is guided in a groove while the upper element of the holder moves downwards and, thus, exerts a shearing force on the sample. The ball bearing mountings at both contact areas of the sample ensure optimum adaptation of the groove holder to the sample geometry.

**Figure 6 micromachines-16-00538-f006:**
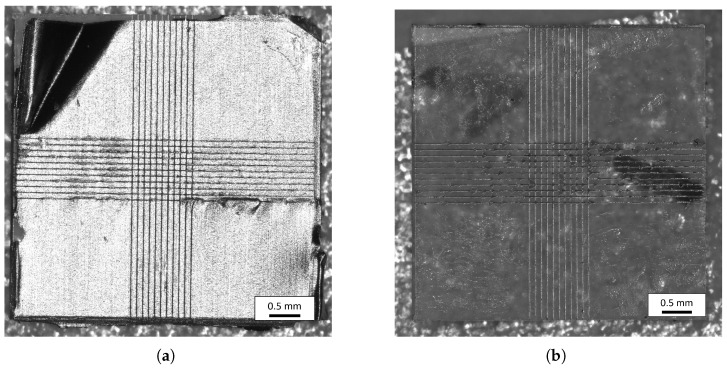
Overview of the samples after the shear strength test was performed. (**a**) shows the glass slide with the metal foil still connected, while (**b**) shows the interface of the second glass sheet, which has been sheared off the aluminum foil.

**Figure 7 micromachines-16-00538-f007:**
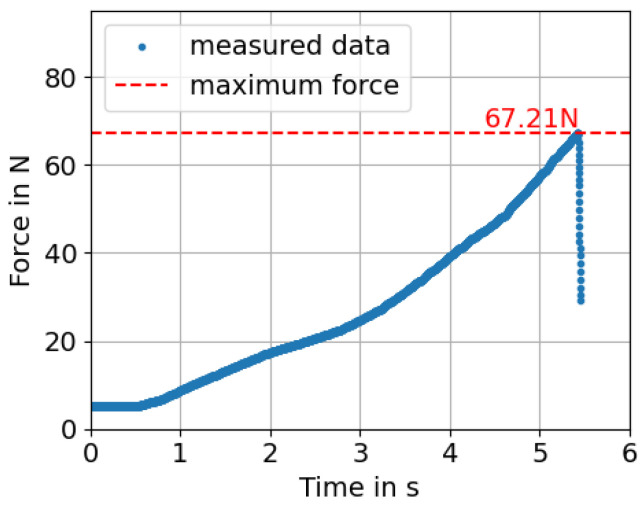
Examplary measurement curve of one sample during the shear test. The measured force is plotted against time. The measurement curve takes the 5.16 N gravitational force of the groove holder into account as an offset. Due to the constant feed rate during the test, the force increases over time until the sample breaks. This can be identified by a sudden drop in the force in the measured data.

**Figure 8 micromachines-16-00538-f008:**
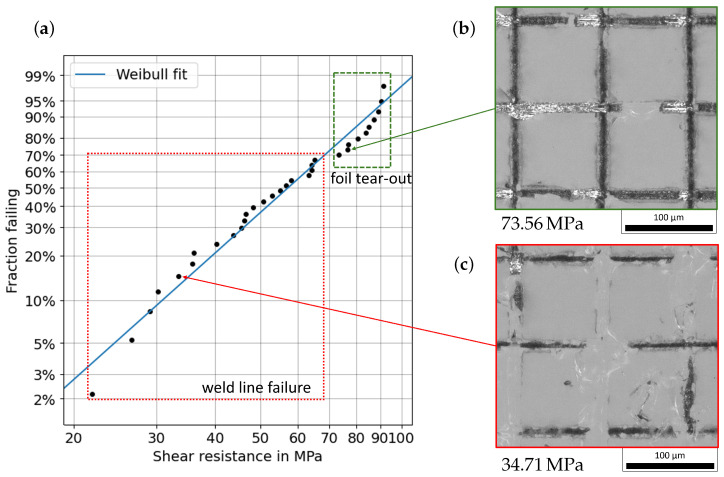
(**a**) Weibull plot of the 32 tested samples. The fraction failing over the shear resistance of the tested samples is shown. The Weibull fit of the data results in an form parameter of α=64.5 and a shape parameter of β=3.1, thus determining the mean shear strength μ of the distribution to be 57.7 MPa. The red and green rectangles cluster the data into the two fracture pattern regimes. In (**b**,**c**), an examplary microscope image of the fracture for each regime is shown. The intermixing zone of the weld seams can be seen as black residual material on the glass slides.

**Table 1 micromachines-16-00538-t001:** Process parameters for the sample preparation.

Pulse energy $E_{\mathrm{pulse}}$	0.43 μJ
Spot diameter $d_{\mathrm{spot}}$	8 μm
Fluence ϕ	$0.86\;J\;cm^{-2}$
Welding speed	$10\;mm\;s^{-1}$
Repetition rate *R*	500 kHz
Number of pulses per burst	4
Temporal separation of pulses within the burst	25 ns

## Data Availability

The data presented in this study are available upon reasonable request from the corresponding authors.

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
