# Peer review of "Ultra-Short-Pulse Laser Welding of Glass to Metal with a Shear Strength Above 50 MPa"

_micromachines, 2025, doi:10.3390/mi16050538_

Round 1
Reviewer 1 Report
Comments and Suggestions for Authors
This research is meaningful because high efficiency laser welding of glass to metal is useful for industry. However, major revision is needed for further improving the writing before publication. Here are some suggestions:
- This article showed the way to improve the efficiency and shear strength for the pulse laser welding of glass to metal. However, the mechanism of how it is achieved should be added. In introduction, why the previous work did not achieve it should also be discussed.
- To make it more readable, the process parameters used in the experiments are suggested to be listed in a table.
- What does "NA" mean in this research? The full form of a term should be written out when introducing an abbreviation for the first time.
- Why did the maximum measured force used in Equation (2) but not the average one or minimum one?
- The mechanism of how this process developed in this research achieved higher efficiency and higher shear strength compared with the previous work should be added and discussed in more details.
- The "mean strength" is mentioned in the conclusion but is not addressed or defined in the main body of the article. This should be revised for clarity and consistency. Please check the whole article to assure there is no similar issue.
Author Response
Comment 1- This article showed the way to improve the efficiency and shear strength for the pulse laser welding of glass to metal. However, the mechanism of how it is achieved should be added. In introduction, why the previous work did not achieve it should also be discussed.
Changes to the manuscript: |
Added the following paragraph to the discussion of the results: (lines 180-192) In contrast to previous work, we did not lapp and polish the metal but used a foil instead, which was pressed between two glass sheets. With this approach we could maximise the contact area. This is not achievable with lapping or polishing, since such a process produces a smooth but bowed surface [Car14]. Indeed, we now assume that the deciding factor for a successful glass to metal bond is not the roughness of the metal surface but solely its waviness. Our approach also allowed for a large number of same quality samples, fundamental to laser parameter studies. With the chosen laser parameters, we obtained weld lines with less cracks or disruptions as well as almost no stress birefringence in their vicinity. However, a detailed investigation of the stress birefringence surrounding the weld line is beyond the scope of this paper. We believe that both factors low waviness and the quality of the weld line contribute to the high shear strength. |
Comment 2- To make it more readable, the process parameters used in the experiments are suggested to be listed in a table.
Changes to the manuscript: |
Added Table 1 with the process parameters to the manuscript |
Comment 3- What does "NA" mean in this research? The full form of a term should be written out when introducing an abbreviation for the first time.
Changes to the manuscript: |
Lines 49-50 The laser beam is focused by a long-distance microscope objective (Mitutoyo) with a numerical aperture (NA) of 0.26,… |
Comment 4- Why did the maximum measured force used in Equation (2) but not the average one or minimum one?
Comment: |
In the case of brittle materials, such as glass, it is common to use a destruction test and to determine the strength of a suitable number of test objects using Weibull statistics. This type of testing requires evaluating the maximum resistance of each sample. We have expanded on the description in the text. |
Changes to the manuscript: |
Lines 139-145 As an example, the measurement curve of one of the 32 samples is plotted in Figure 7. During the test, the shear force increases over time, because of the constant feed rate. After the maximum shear force of the sample is reached, the sample breaks apart. For this sample the maximum force is . By dividing the force with the welding area the shear strength of this sample determines to (3) This destructive shear test was performed on all 32 samples and the shear strength of each sample was determined using Equation 3. |
Comment 5- The mechanism of how this process developed in this research achieved higher efficiency and higher shear strength compared with the previous work should be added and discussed in more details.
Changes to the manuscript: |
Added the following paragraph to the discussion of the results: (lines 180-192) In contrast to previous work, we did not lapp and polish the metal but used a foil instead, which was pressed between two glass sheets. With this approach we could maximise the contact area. This is not achievable with lapping or polishing, since such a process produces a smooth but bowed surface [Car14]. Indeed, we now assume that the deciding factor for a successful glass to metal bond is not the roughness of the metal surface but solely its waviness. Our approach also allowed for a large number of same quality samples, fundamental to laser parameter studies. With the chosen laser parameters, we obtained weld lines with less cracks or disruptions as well as almost no stress birefringence in their vicinity. However, a detailed investigation of the stress birefringence surrounding the weld line is beyond the scope of this paper. We believe that both factors low waviness and the quality of the weld line contribute to the high shear strength. |
Comment 6- The "mean strength" is mentioned in the conclusion but is not addressed or defined in the main body of the article. This should be revised for clarity and consistency. Please check the whole article to assure there is no similar issue.
Comment: |
Mean value and expectation value of the Weibull distribution are the same. We change the text to mean value in all places to avoid confusion. |
Changes to the manuscript: |
P7, Line 149: Therefore, the mean value μ of the Weibull-distribution is P8: Figure 8: which determines the mean shear strength μ of the distribution to 57.7MPa |

Reviewer 2 Report
Comments and Suggestions for Authors
The paper reports on ultrashort pulse laser welding of Borofloat glass to aluminum, achieving a weld seam with a shear strength exceeding 50 MPa. Additionally, the microstructure of the weld was analyzed using SEM and EDX methods. The topic aligns well with the scope of the journal.
However, I have several remarks regarding the content:
- The authors mention that “the laser was set to 500 kHz with a pulse train (burst) of four sub-pulses, separated by 25 ns.” What theoretical or experimental justification was used to select this specific burst regime? Please clarify the rationale behind this choice.
- The manuscript does not provide key laser parameters such as fluence, beam diameter, or the beam waist position within the glass. Including these values is essential to allow readers to understand and potentially reproduce the welding process.
- There is no discussion regarding the mechanism of ultrashort pulse laser welding in this specific glass-metal interface. A brief description would strengthen the scientific foundation of the study.
After minor revisions the paper can be published.
Author Response
Comments 1- The authors mention that “the laser was set to 500 kHz with a pulse train (burst) of four sub-pulses, separated by 25 ns.” What theoretical or experimental justification was used to select this specific burst regime? Please clarify the rationale behind this choice.
Changes to the manuscript: |
Added sentences in line 69-72: In order to operate in the heat accumulation regime during the welding process the repetition rate R of the USP laser was set to 500 kHz. Motivated by the studies of Richter et al. [10] and Hecker et al.[11] we decided to use a pulse burst to reduce cracks and weld line disruptions in the glass. Each pulse is split in to four sub pulses, separated by 25 ns. Figure 2 shows a schematic of the burst mode used during the welding process. Added figure 2 to clarify the pulse regime |
Comments 2- The manuscript does not provide key laser parameters such as fluence, beam diameter, or the beam waist position within the glass. Including these values is essential to allow readers to understand and potentially reproduce the welding process.
Changes to the manuscript: |
Lines 45-46: The 1/e2 beam diameter of the unfocused laser beam is 3.3 mm
Lines 75-77 The glass metal interface is in line with the focal plane of the Objective, resulting in a fluence of at the interface. The process parameters are summarized in Table 1.
|
Comments 3- There is no discussion regarding the mechanism of ultrashort pulse laser welding in this specific glass-metal interface. A brief description would strengthen the scientific foundation of the study.
Comment: |
Localized heating in the interface of both metal and glass, forming of a liquid zone, where components of glass and metal mix. However, a detailed discussion requires more in detail measurements of the weld seams, which is beyond the scope of this paper. |
Changes to the manuscript: |
Lines:40-41 These laser conditions were chosen to form a liquid zone, localized at the metal-glass interface where both components would mix. Line 69: In order to operate in the heat accumulation regime during the welding process.. Added the following paragraph to the discussion of the results: (lines 180-192) In contrast to previous work, we did not lapp and polish the metal but used a foil instead, which was pressed between two glass sheets. With this approach we could maximise the contact area. This is not achievable with lapping or polishing, since such a process produces a smooth but bowed surface [Car14]. Indeed, we now assume that the deciding factor for a successful glass to metal bond is not the roughness of the metal surface but solely its waviness. Our approach also allowed for a large number of same quality samples, fundamental to laser parameter studies. With the chosen laser parameters, we obtained weld lines with less cracks or disruptions as well as almost no stress birefringence in their vicinity. However, a detailed investigation of the stress birefringence surrounding the weld line is beyond the scope of this paper. We believe that both factors low waviness and the quality of the weld line contribute to the high shear strength. |

Round 2
Reviewer 1 Report
Comments and Suggestions for Authors
I believe the manuscript has been well revised and meet the requirements of the Journal of Micromachines. Therefore I recommend it for publication in current form.